# A Pilot Study of the Computerized Brief Smell Identification Test

**DOI:** 10.3390/diagnostics14192121

**Published:** 2024-09-25

**Authors:** Rong-San Jiang, Yi-Fang Chiang

**Affiliations:** 1Department of Otolaryngology, Taichung Veterans General Hospital, Taichung 40705, Taiwan; rsjiang@vghtc.gov.tw; 2Department of Medical Research, Taichung Veterans General Hospital, Taichung 40705, Taiwan; 3School of Medicine, Chung Shan Medical University, Taichung 40201, Taiwan; 4School of Medicine, College of Medicine, National Yang Ming Chiao Tung University, Taipei 30010, Taiwan

**Keywords:** Brief Smell Identification Test, Computerized Brief Smell Identification Test, olfactory identification test, olfactory dysfunction, test–retest

## Abstract

**Objectives:** A self-administered Computerized Brief Smell Identification Test (cB-SIT) was developed recently to perform the olfactory identification test under computer control. The aim of this study was to evaluate the clinical applicability of the cB-SIT as compared with the traditional Brief Smell Identification Test (B-SIT). **Methods:** Sixty healthy volunteers with self-reported normal olfactory function, 30 hyposmic patients, and 30 anosmic patients were enrolled from June 2023 to May 2024. All enrolled participants received both B-SIT and cB-SIT in a random order to measure their odor identification ability. Thirty healthy volunteers took the second B-SIT and cB-SIT at least one week later. **Results:** The score was significantly different in both B-SIT and cB-SIT among healthy volunteers, hyposmic, and anosmic patients. The correct answer rate was significantly different in 10 items of the B-SIT and in 7 items of the cB-SIT among the three groups, but the post hoc test showed significant differences in correct answer rates between healthy volunteers and hyposmic patients in 7 items of both the B-SIT and cB-SIT. Test–retest results showed the score of the second B-SIT test was significantly higher than that of the first test, but the scores of the two tests of the cB-SIT were not significantly different. In the B-SIT, the lemon odorant had a higher correct answer rate in the second test than in the first test, but in the cB-SIT, the correct answer rate was not significantly different between the first and second tests in all 12 items. **Conclusions:** Our findings demonstrate that the cB-SIT was similar to the B-SIT and can be administered in the diagnosis of patients with olfactory dysfunction.

## 1. Introduction

The function of the olfactory system allows people to detect food and hazards. Olfactory dysfunction results in a substantial adverse influence on quality of life. Loss of olfactory function is also a prodromal marker for Parkinson’s disease and is implicated in a range of other neurological and psychiatric disorders including Alzheimer’s disease, schizophrenia, and major depression [1]. However, people are generally unaware of olfactory loss until formal testing, highlighting the necessity of a subjective measurement of olfactory function.

A comprehensive smell test is comprised of an odor threshold test, an odor discrimination test and, an odor identification test [2]. The odor identification test is more sensitive than the odor threshold test or odor discrimination test to detect Parkinson’s and Alzheimer’s diseases [3]. The self-administered University of Pennsylvania Smell Identification Test (UPSIT), which consists of 40 individual tests, is one of the most commonly used odor identification tests [4]. However, a few of its items or responses are unfamiliar to non–North Americans [5]. Therefore, the same group developed a 12-item odor identification test based upon items from the UPSIT [6]. The Cross-Cultural Smell Identification Test, which is also known as the Brief Smell Identification Test (B-SIT), was developed and contains 12 UPSIT items that are familiar to most persons from North American, European, South American, and Asian cultures [7].

The B-SIT employs the following odorants: cinnamon, turpentine, lemon, smoke, chocolate, rose, paint thinner, banana, pineapple, gasoline, soap, and onion [8]. It is contained in a single booklet. The odorants are embedded in 10–50 μm microcapsules fixed in a proprietary binder and positioned on scent strips on the bottom of the pages of the test booklets. To conduct the test, the examiner uses an enveloped pencil to scratch the odor strip, releasing the odorant. Thereafter, the strip is placed beneath the subject’s nose, and the subject is asked to identify the odor from a set of four descriptors. A response is required for each item even if no odor is perceived [9]. The test score is determined by the number of correctly identified odors and can range from 0 to 12 [10]. It can be self-administered in less than 5 min [7]. Currently, the B-SIT is widely used to assess olfactory function in multicultural countries [11,12,13].

A computer-controlled olfactometer for a self-administered odor identification test has been designed to save time and cost in clinical practice [14]. A benefit of this computer-controlled olfactometer compared with traditional self-administered odor identification tests, is that the odors are not contaminated by inadequate operation by the examinees [15]. It employs 10 odorants: pear, rose, apple, cinnamon, peppermint, coffee, chocolate, cola, melon, and orange [14]. In a study, it was concluded that it was appropriate for a self-administered odor identification test based on its validity and test–retest reliability [14]. Recently, a new version of the Self-administered Computerized Olfactory Testing System (SCOTS) has been developed. It can perform both the threshold test and the Computerized Brief Smell Identification Test (cB-SIT). In our previous study, the SCOTS was appropriate for a self-administered threshold test based on its validity and test–retest reliability [16]. The SCOTS has several benefits. It can provide exact control of the stimulus duration and inter-stimulus intervals [17]. The purpose of this study was to evaluate the clinical applicability of the cB-SIT.

## 2. Materials and Methods

### 2.1. Population

Sixty healthy volunteers with self-reported normal olfactory function, 30 hyposmic patients, and 30 anosmic patients were enrolled from June 2023 to May 2024. The exclusion criteria included any healthy volunteers who had been diagnosed with chronic rhinosinusitis or allergic rhinitis, had had an episode of upper respiratory infection within one week before the test, or who were pregnant or lactating. The hyposmic patient group comprised individuals who complained of olfactory dysfunction and had an olfactory threshold between −1 and −6 using the phenyl ethyl alcohol (PEA) odor detection threshold test. The anosmic patient group consisted of individuals who complained of complete loss of olfactory function and had a PEA threshold of −1.

All enrolled participants received both the B-SIT and SCOTS–based cB-SIT in a random order to measure their odor identification ability. Thirty healthy volunteers took the second B-SIT and cB-SIT at least one week later. This study was approved by the Institutional Review Board (II) of Taichung Veterans General Hospital (TCVGH IRB No. CE23131B). Written consent was obtained from each patient.

### 2.2. Phenyl Ethyl Alcohol Odor Detection Threshold Test

In the PEA odor detection threshold test [16,18], a two-alternative forced-choice single-staircase detection threshold procedure is performed. Firstly, two glass sniff bottles are presented to the subject. One contains 20 mL of a given concentration of PEA dissolved in light mineral oil, whereas the other contains the mineral oil alone. These two bottles are opened and placed under the subject’s nose in a random order. The subject indicates which bottle contains the stronger odor. If no difference is perceived, a guess is needed. 

The PEA concentrations ranged from 10^−1^ to 10^−9^ log *v*/*v* in half-log concentration steps. The test begins with the bottle containing PEA odorant at 10^−6^ log vol/vol. Correct identification of the bottle that contains the PEA odorant at 10^−6^ log *v*/*v* in five successive trials triggers a reversal of the staircase to the next lower concentration, whereas a single incorrect identification triggers the reversal of the staircase to the next higher concentration. In the following, correct identification of the bottle that contains the PEA odorant in two successive trials trigger a reversal of the staircase to the next lower concentration. The test is completed with seven reversals. The geometric mean of the last four reversed points of the seven reversals is used as the threshold estimate.

### 2.3. Brief Smell Identification Test

The B-SIT (Sensonics, Inc., Haddon Heights, NJ, USA) consists of 12 items (Figure 1). The odorants in these items are cinnamon, turpentine, lemon, smoke, chocolate, rose, paint thinner, banana, pineapple, gasoline, soap, and onion in descending order (Table 1) [19]. They are all chosen from the odorants used in the UPSIT [20]. However, 8 of the 12 sets of descriptors used to identify the smelled odor are different between the B-SIT and UPSIT.

The test booklet used in the B-SIT is similar to that used in the UPSIT. Each of the 12 ‘scratch & sniff’ odorants are embedded in 10 to 50 µm urea-formaldehyde polymer microcapsules fixed in a propriety binder and positioned on brown strips located at the bottom of the pages of each test booklet. The testing procedures of the B-SIT are the same as those used in the UPSIT [21]. When the subject takes the UPSIT, he/she releases each of the 12 odorants by scratching the strip with a pencil tip in a standardized manner. The identity of the released odorant is signified by choosing a name from a set of 4 odor descriptors. The test is scored as the number of odors identified correctly. A response is required for each test item even if no smell is perceived (i.e., the test is forced choice), allowing for the detection of malingering on the basis of improbable responses.

### 2.4. Computerized Brief Smell Identification Test

A 40CH-RA SCOTS (Sensonics, Inc., Haddon Heights, NJ, USA) was used to perform the cB-SIT (Figure 2). It also consists of 12 items and is a forced-choice odor identification test, as with the B-SIT. However, the scents in these items are cinnamon, rubber tire, lemon, smoke, chocolate, rose, licorice, banana, pineapple, apple, soap, and onion in descending order (Table 1). They are all used in the UPSIT, except apple. However, 8 of the 11 sets of descriptors used to identify the smelled odor are different between the cB-SIT and UPSIT. Among the 12 items in the B-SIT and cB-SIT, 9 tested odorants with their sets of descriptors are the same, but the released odorants are different for the second, seventh, and tenth items (Table 1).

To begin the cB-SIT test, the testing procedures were initiated by opening the Sensonics Olfactometer program. According to the instructions shown on the touch screen, the subject conducted the test at his/her own pace. The subject clicks on the cB-SIT button on the main screen of the computer. Then, the subject is prompted to click another button to begin the test. Once the test begins, a scent is released. The subject selects an answer from a set of 4 descriptors that he/she believes most closely resembles the scent within 30 s. There is a pause of 7 s before proceeding to the next item. The steps are repeated until all 12 items are completed. Our subjects typically completed the test in less than 5 min.

### 2.5. Statistical Analysis

Descriptive data are presented as means ± standard deviations. Our data were not normally distributed. The gender ratio and the correct answer rate of each item in the B-SIT and cB-SIT were compared among healthy volunteers, hyposmic patients, and anosmic patients using Pearson’s chi-squared test. The age and the scores of the B-SIT and cB-SIT of these 3 groups of subjects were compared using the Kruskal–Wallis test. The correct answer rate of each item was compared between the B-SIT and cB-SIT using the McNemar test. The scores of the B-SIT and cB-SIT were compared between the first and second tests in 30 healthy volunteers using the Wilcoxon Signed Ranks test. All computations were performed with SPSS (version 22.0, SPSS, Inc., Chicago, IL, USA). Two-tailed *p*-values < 0.05 were considered statistically significant.

## 3. Results

### 3.1. Patients

There were 19 males and 43 females in the group of healthy volunteers, 12 males and 18 females in the hyposmic group, and 16 males and 14 females in the anosmic group. The gender ratio was not significantly different among the three groups (*p* = 0.066). The ages ranged from 22 to 80 years old, with a mean of 43.1 ± 17.3 years, in the group of healthy volunteers, from 18 to 68 years old, with a mean of 45.2 ± 13.5 years, in the hyposmic group, and from 22 to 71 years old, with a mean of 50.5 ± 12.7 years, in the anosmic group. The ages of the subjects were not significantly different among the three groups (*p* = 0.063).

### 3.2. Test–Retest Reliability 

Table 1 shows a comparison of the score and correct answer rate of each item of the B-SIT and cB-SIT between the first test and second test among 30 healthy volunteers. The score of the B-SIT was significantly higher in the second test than in the first test, but the score of the cB-SIT was not significantly different between the first and second tests. In the B-SIT, the correct answer rate of the lemon odorant was significantly higher in the second test than in the first test, but in the cB-SIT, the correct answer rates of all 12 odorants were not significantly different between the first and second tests.

### 3.3. Healthy Group

Table 2 shows the score and correct answer rate of each item of the B-SIT and cB-SIT in the group of healthy volunteers. The correct answer rate of each item was the number of healthy volunteers who correctly identified the odorant of that item divided by the 60 healthy volunteers. The score ranged from 7 to 12 with a mean of 9.3 ± 1.5 in the B-SIT and from 4 to 12 with a mean of 8.6 ± 1.7 in the cB-SIT.

The score of the B-SIT was significantly higher than that of the cB-SIT among 60 healthy volunteers. In the B-SIT, more than 90% of the subjects correctly identified six odorants: turpentine, chocolate, paint thinner, pineapple, soap, and onion. However, only about half of the subjects correctly identified four odorants: lemon, smoke, rose, and banana. Furthermore, in the cB-SIT, more than 90% of subjects correctly identified four odorants: licorice, pineapple, soap, and onion. However, fewer than half of the subjects correctly identified three odorants: lemon, chocolate, and banana. A comparison of the correct answer rate between the B-SIT and cB-SIT revealed that significantly more subjects correctly identified the chocolate and banana odorants in the B-SIT than in the cB-SIT, but significantly more subjects correctly identified the smoke and rose odorants in the cB-SIT than in the B-SIT. Among the three odorants that were different between the B-SIT and cB-SIT, the correct answer rate was not significantly different between turpentine in the B-SIT and rubber tire in the cB-SIT and between paint thinner in the B-SIT and licorice in the cB-SIT. However, significantly more subjects correctly identified the gasoline odorant in the B-SIT than the apple odorant in the cB-SIT.

### 3.4. Hyposmic Patients

Table 2 shows the score and correct answer rate of each item of the B-SIT and cB-SIT in the hyposmic group of patients. The correct answer rate of each item was the number of hyposmic patients who correctly identified the odorant of that item divided by the 30 hyposmic patients. The score ranged from 1 to 11 with a mean of 6.5 ± 2.5 in the B-SIT and from 2 to 9 with a mean of 5.6 ± 1.9 in the cB-SIT.

The score of the B-SIT was significantly higher than that of the cB-SIT among the 30 hyposmic patients. A comparison of the correct answer rate between the B-SIT and cB-SIT revealed that significantly more subjects correctly identified the chocolate odorant in the B-SIT than in the cB-SIT. Among the three odorants that were different between the B-SIT and cB-SIT, the correct answer rate was not significantly different between turpentine in the B-SIT and rubber tire in the cB-SIT and between gasoline in the B-SIT and apple in the cB-SIT. However, significantly more subjects correctly identified the paint thinner odorant in the B-SIT than the licorice odorant in the cB-SIT.

### 3.5. Anosmic Patients 

Table 2 shows the score and correct answer rate of each item of the B-SIT and cB-SIT in the group of anosmic patients. The correct answer rate of each item was the number of anosmic patients who correctly identified the odorant of that item divided by the 30 anosmic patients. The score ranged from 1 to 9 with a mean of 4.3 ± 2.3 in the B-SIT and from 1 to 6 with a mean of 4.5 ± 1.4 in the cB-SIT.

The score was not significantly different between the B-SIT and cB-SIT among the 30 anosmic patients. A comparison of the correct answer rate between the B-SIT and cB-SIT revealed that there were no significant differences in all 12 items, including the three odorants that were different between the B-SIT and cB-SIT.

### 3.6. Comparison among Healthy Volunteers, Hyposmic Patients, and Anosmic Patients 

Table 2 shows the comparison of the score and correct answer rate of each item of the B-SIT and cB-SIT among healthy volunteers, hyposmic patients, and anosmic patients. 

The score was significantly different in either the B-SIT or cB-SIT among the three groups. The post hoc test showed significant differences in scores between healthy volunteers and hyposmic patients (*p* < 0.001), between healthy volunteers and anosmic patients (*p* < 0.001), and between hyposmic and anosmic patients (*p* = 0.045) in the B-SIT. On the other hand, in the cB-SIT, the post hoc test showed significant differences in scores between healthy volunteers and hyposmic patients (*p* < 0.001) and between healthy volunteers and anosmic patients (*p* < 0.001) but not between hyposmic and anosmic patients (*p* = 0.290).

The correct answer rate was significantly different in 10 items of the B-SIT among the three groups except lemon and banana. The post hoc test showed significant differences in the correct answer rates of seven items between healthy volunteers and hyposmic patients, including cinnamon (*p* < 0.001), turpentine (*p* < 0.001), paint thinner (*p* = 0.014), pineapple (*p* < 0.001), gasoline (*p* = 0.020), soap (*p* < 0.001), and onion (*p* = 0.008). The post hoc test showed significant differences in the correct answer rates of 10 items between healthy volunteers and anosmic patients, including cinnamon (*p* < 0.001), turpentine (*p* < 0.001), smoke (*p* = 0.002), chocolate (*p* < 0.001), rose (*p* = 0.006), paint thinner (*p* < 0.001), pineapple (*p* < 0.001), gasoline (*p* = 0.002), soap (*p* < 0.001), and onion (*p* < 0.001). The post hoc test showed significant differences in the correct answer rates of two items between hyposmic and anosmic patients, including chocolate (*p* = 0.011) and paint thinner (*p* = 0.002).

The correct answer rate was significantly different in seven items of the cB-SIT among the three groups except rubber tire, lemon, chocolate, banana, and apple. The post hoc test showed significant differences in the correct answer rates of seven items between healthy volunteers and hyposmic patients, including cinnamon (*p* < 0.001), smoke (*p* < 0.001), rose (*p* < 0.001), licorice (*p* < 0.001), pineapple (*p* < 0.001), soap (*p* = 0.026), and onion (*p* = 0.041). The post hoc test showed significant differences in the correct answer rates of seven items between healthy volunteers and anosmic patients, including cinnamon (*p* < 0.001), smoke (*p* < 0.001), rose (*p* < 0.001), licorice (*p* < 0.001), pineapple (*p* < 0.001), soap (*p* < 0.001), and onion (*p* < 0.001). The post hoc test showed a significant difference in the correct answer rate for onion (*p* = 0.003) between hyposmic and anosmic patients.

## 4. Discussion

The B-SIT is a commercially available smell identification test. Its norms have been established and its results are comparable with those of other olfactory tests [22]. Therefore, in this study we compared the olfactory performance of the cB-SIT with that of the B-SIT. Both tests use 12 odorants to measure smell identification ability, although the cB-SIT has 3 odorants that are different from those used in the B-SIT. The testing procedures are similar between the B-SIT and cB-SIT. They differ only in the manner of odorant presentation.

The score of the B-SIT was significantly higher than that of the cB-SIT in the healthy and hyposmic groups. They had more subjects who correctly identified chocolate and banana odorants in the B-SIT than the same odorants in the cB-SIT. One possible reason for this might be that no time limit was set for subjects to smell the released odorants in the B-SIT [23]. The time limit was set at 30 s for the cB-SIT in this study. In addition, a higher number of subjects correctly identified smoke and rose odorants in the cB-SIT than in the B-SIT. Among the three odorants that are different between the B-SIT and cB-SIT, the correct answer rate was not significantly different between turpentine in the B-SIT and rubber tire in cB-SIT and between paint thinner in the B-SIT and licorice in the cB-SIT. However, significantly more subjects correctly identified the gasoline odorant in the B-SIT than the apple odorant in the cB-SIT.

The B-SIT has been shown to be a valid test for assessing olfactory function in many disorders, including Parkinson’s disease and COVID-19-related olfactory dysfunction [24,25,26]. In this study, the score was significantly different in both the B-SIT and cB-SIT among the three groups, although the post hoc test did not show a significant difference between the hyposmic and anosmic patients in the cB-SIT. The correct answer rate was significantly different in 10 items of the B-SIT and in 7 items of the cB-SIT among the three groups, but the post hoc test showed significant differences in the correct answer rates between healthy volunteers and hyposmic patients in eight items of the B-SIT and in seven items of the cB-SIT. This indicates that the cB-SIT had a similar ability to that of B-SIT in differentiating subjects with normal olfactory function from patients with olfactory dysfunction.

The test–retest results showed the score of the second test of the B-SIT was significantly higher than that of the first test, but the scores of the two tests of the cB-SIT were not significantly different. In the B-SIT, the lemon odorant had a higher correct answer rate in the second test than the first test, but in the cB-SIT, the correct answer rate was not significantly different between the first and second tests in all 12 items. The results indicate that the cB-SIT had better test–retest reliability than the B-SIT.

## 5. Conclusions

Our results showed the test score was significantly different in both the B-SIT and cB-SIT among healthy volunteers, hyposmic patients, and anosmic patients. The correct answer rate was significantly different in 10 items of the B-SIT and in 7 items of the cB-SIT among the three groups, but the post hoc test showed significant differences in correct answer rates between healthy volunteers and hyposmic patients in seven items of both the B-SIT and the cB-SIT. The test–retest results showed the score of the second test of the B-SIT was significantly higher than that of the first test, but the scores of the two tests of the cB-SIT were not significantly different. These demonstrate that the cB-SIT is similar to the B-SIT and can be administered in the diagnosis of patients with olfactory dysfunction. It has the advantages of saving time and cost in clinical practice.

## Figures and Tables

**Figure 1 diagnostics-14-02121-f001:**
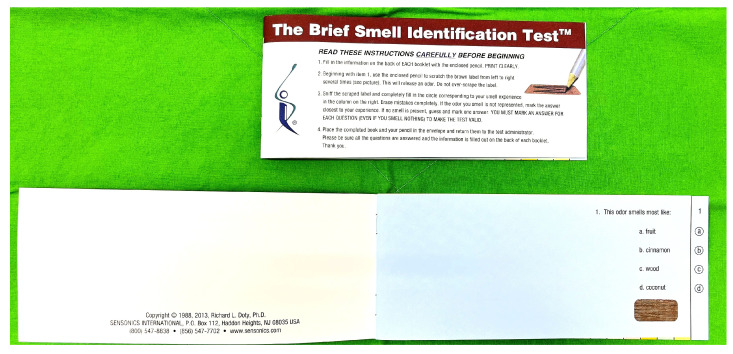
Brief Smell Identification Test.

**Figure 2 diagnostics-14-02121-f002:**
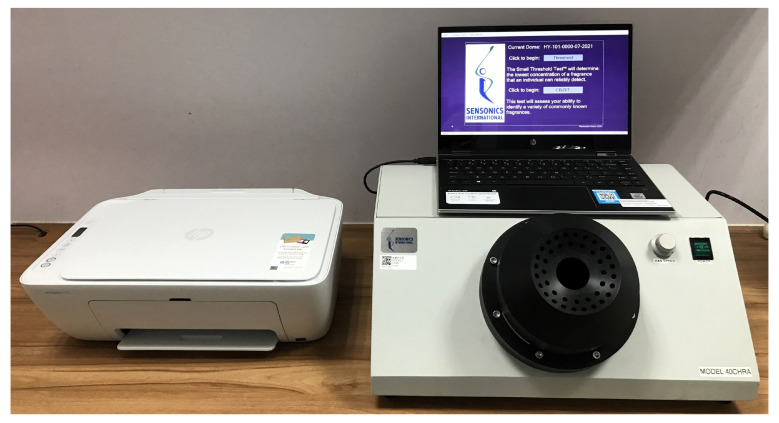
Computerized Brief Smell Identification Test.

**Table 1 diagnostics-14-02121-t001:** Odorants used in Brief Smell Identification Test (B-SIT) and Computerized Brief Identification Test (cB-SIT) and test–retest reliability.

Item	B-SIT Score	First Test	Second Test	*p*-Value	cB-SIT Score	First Test	Second Test	*p*-Value
Score		9.3 ± 1.4	10.4 ± 1.0	0.003		9.1 ± 1.3	9.1 ± 1.5	0.717
1. Cinnamon		63.3%	60.0%	1.000		93.3%	76.7%	0.125
2. Turpentine		90.0%	86.7%	1.000	Rubber tire	90.0%	86.7%	1.000
3. Lemon		46.7%	80.0%	0.013		50.0%	53.3%	1.000
4. Smoke		66.7%	86.7%	0.070		90.0%	90.0%	1.000
5. Chocolate		93.3%	100%	0.500		23.3%	20.0%	1.000
6. Rose		56.7%	80.0%	0.065		90.0%	90.0%	1.000
7. Paint thinner		96.7%	100%	1.000	Licorice	86.7%	86.7%	1.000
8. Banana		43.3%	63.3%	0.109		23.3%	33.3%	0.375
9. Pineapple		96.7%	96.7%	1.000		96.7%	100%	1.000
10. Gasoline		83.3%	86.7%	1.000	Apple	70.0%	76.7%	0.754
11. Soap		90.0%	96.7%	0.500		96.7%	100%	1.000
12. Onion		100%	100%			96.7%	100%	1.000

**Table 2 diagnostics-14-02121-t002:** Comparison of the score and correct answer rate of each item among the 3 groups and between the Brief Smell Identification Test (B-SIT) and Computerized Brief Identification Test (cB-SIT).

Item	Score Type	Healthy Volunteers	Hyposmic Patients	Anosmic Patients	*p*
Score	B-SIT	9.3 ± 1.5	6.5 ± 2.5	4.3 ± 2.3	<0.001 ^a^
	cB-SIT	8.6 ± 1.7	5.6 ± 1.9	4.5 ± 1.4	<0.001 ^a^
	*p*	0.008 ^b^	0.04 ^b^	0.435	
Cinnamon	B-SIT	71.7%	23.3%	10.0%	<0.001 ^a^
	cB-SIT	80.5%	33.3%	23.3%	<0.001 ^a^
	*p*	0.077	0.508	0.289	
Turpentine	B-SIT	93.3%	56.7%	43.3%	<0.001 ^a^
Rubber tire	cB-SIT	83.3%	63.3%	66.7%	0.069
	*p*	0.180	0.774	0.167	
Lemon	B-SIT	51.7%	40.0%	33.3%	0.222
	cB-SIT	45.0%	26.7%	23.3%	0.069
	*p*	0.523	0.344	0.549	
Smoke	B-SIT	55.0%	33.3%	16.7%	0.002 ^a^
	cB-SIT	85.0%	36.7%	33.3%	<0.001 ^a^
	*p*	<0.001 ^a^	1.000	0.227	
Chocolate	B-SIT	90.0%	80.0%	43.3%	<0.001 ^a^
	cB-SIT	20.0%	40.0%	36.7%	0.084
	*p*	<0.001 ^a^	0.004 ^a^	0.804	
Rose	B-SIT	58.3%	33.3%	23.3%	0.003 ^a^
	cB-SIT	85.0%	30.0%	16.7%	<0.001 ^a^
	*p*	<0.001 ^a^	1.000	0.727	
Paint thinner	B-SIT	95.0%	76.7%	33.3%	<0.001 ^a^
Licorice	cB-SIT	91.7%	50.0%	43.3%	<0.001 ^a^
	*p*	0.727	0.039 ^b^	0.607	
Banana	B-SIT	50.0%	53.3%	33.3%	0.229
	cB-SIT	20.0%	30.0%	13.3%	0.276
	*p*	0.001 ^a^	0.118	0.070	
Pineapple	B-SIT	91.7%	53.3%	40.0%	<0.001 ^a^
	cB-SIT	90.0%	43.3%	40.0%	<0.001 ^a^
	*p*	1.000	0.581	1.000	
Gasoline	B-SIT	80.0%	53.3%	43.3%	0.001 ^a^
Apple	cB-SIT	58.3%	46.7%	56.7%	0.564
	*p*	0.015 ^b^	0.815	0.388	
Soap	B-SIT	95.0%	63.3%	53.3%	<0.001 ^a^
	cB-SIT	93.3%	73.3%	50.0%	<0.001 ^a^
	*p*	1.000	0.581	1.000	
Onion	B-SIT	98.3%	80.0%	53.3%	<0.001 ^a^
	cB-SIT	98.3%	86.7%	46.7%	<0.001 ^a^
	*p*	1.000	0.625	0.727	

^a^: *p* < 0.005; ^b^: *p* < 0.05.

## Data Availability

The data presented in this study are available on request from the corresponding author.

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
