# Peer review of "A Pilot Study of the Computerized Brief Smell Identification Test"

_diagnostics, 2024, doi:10.3390/diagnostics14192121_

Round 1
Reviewer 1 Report
Comments and Suggestions for Authors
General considerations on the manuscript: The manuscript presents a thorough and well-structured analysis of olfactory testing methodologies, focusing on the comparison between the Computerized Brief Smell Identification Test (cB-SIT) and the traditional Brief Smell Identification Test (B-SIT). The study design is robust, incorporating a diverse sample of participants, including healthy volunteers, hyposmic patients, and anosmic patients. The inclusion of multiple assessments and a one-week interval between tests strengthens the reliability of the findings. Overall, the manuscript is original and provides valuable insights into the clinical applicability of the cB-SIT. It effectively addresses the need for a reliable, self-administered olfactory test with minimal human error. The results suggest that the cB-SIT is comparable to the B-SIT, making it a viable alternative for diagnosing olfactory dysfunction. The manuscript is well-written and does not exhibit significant errors. However, attention to minor typographical details, such as consistent capitalization of the "P" in "significance" across tables and proper alignment of values, would enhance its clarity and professionalism. These adjustments will ensure that the presentation of data is precise and easily interpretable for readers.
Specific comments: 1. The sentence "A comprehensive smell test…..test and, odor identification test." should include a bibliographic reference, such as: Hummel, Thomas, et al. "'Sniffin'sticks': olfactory performance assessed by the combined testing of odor identification, odor discrimination and olfactory threshold." Chemical Senses 22.1 (1997): 39-52.
2. The sentence "Any healthy volunteers who had been diagnosed with chronic rhinosinusitis …..were pregnant or lactating were excluded." could be rewritten as follows to be clearer for readers: The exclusion criteria included any healthy volunteers who had been diagnosed with chronic rhinosinusitis or allergic rhinitis, had an episode of upper respiratory infection within one week before the test, or were pregnant or lactating.
Author Response
General considerations on the manuscript: The manuscript presents a thorough and well-structured analysis of olfactory testing methodologies, focusing on the comparison between the Computerized Brief Smell Identification Test (cB-SIT) and the traditional Brief Smell Identification Test (B-SIT). The study design is robust, incorporating a diverse sample of participants, including healthy volunteers, hyposmic patients, and anosmic patients. The inclusion of multiple assessments and a one-week interval between tests strengthens the reliability of the findings. Overall, the manuscript is original and provides valuable insights into the clinical applicability of the cB-SIT. It effectively addresses the need for a reliable, self-administered olfactory test with minimal human error. The results suggest that the cB-SIT is comparable to the B-SIT, making it a viable alternative for diagnosing olfactory dysfunction. The manuscript is well-written and does not exhibit significant errors. However, attention to minor typographical details, such as consistent capitalization of the "P" in "significance" across tables and proper alignment of values, would enhance its clarity and professionalism. These adjustments will ensure that the presentation of data is precise and easily interpretable for readers.
Answer: Thanks for your comments. Yes, we corrected typographical errors and alignment errors of values in Tables.
Specific comments: 1. The sentence "A comprehensive smell test…..test and, odor identification test." should include a bibliographic reference, such as: Hummel, Thomas, et al. "'Sniffin'sticks': olfactory performance assessed by the combined testing of odor identification, odor discrimination and olfactory threshold." Chemical Senses 22.1 (1997): 39-52.
Answer: Thanks for your suggestion. Yes, we have added this in our references.
- The sentence "Any healthy volunteers who had been diagnosed with chronic rhinosinusitis …..were pregnant or lactating were excluded." could be rewritten as follows to be clearer for readers: The exclusion criteria included any healthy volunteers who had been diagnosed with chronic rhinosinusitis or allergic rhinitis, had an episode of upper respiratory infection within one week before the test, or were pregnant or lactating.
Answer: Thanks for your suggestion. Yes, we have changed the sentence as you recommended.
Reviewer 2 Report
Comments and Suggestions for Authors
The Manuscript (ID: diagnostics-3183757) entitled “A Pilot Study of the Computerized Brief Smell Identification Test” evaluates an interesting topic. The aim of this study was to evaluate the clinical applicability of the Computerized Brief Smell Identification Test (cB-SIT) as compared with the traditional Brief Smell Identification Test (B-SIT). The authors suggested that the cB-SIT was similar to the B-SIT and could be administered in the diagnosis of patients with olfactory dysfunction.
In general, the Manuscript is well written and clear to understand, consequently it requires some major revisions.
Specific comments:
The enrollment of healthy subjects should be performed considering a clinical evaluation, since self-reported may be different from the medical evaluation.
In the patient section, it should be “had an …" and also it should be "threshold of ...".
The authors should describe how they decided on the concentration used for PEA test. Please indicate a reference for this test.
In the statistical analyses, authors should describe if data are normal distributed.
In the Table 2, authors should use alphabetical letter to indica the significance between groups.
In addition, a clear legend should be indicated in Table 2.
The section of test-retest reliability should be indicated in the first section of the results, since all tables descriptions should be in numerical order.
Comments on the Quality of English LanguageMinor editing of English language is required
Author Response
The Manuscript (ID: diagnostics-3183757) entitled “A Pilot Study of the Computerized Brief Smell Identification Test” evaluates an interesting topic. The aim of this study was to evaluate the clinical applicability of the Computerized Brief Smell Identification Test (cB-SIT) as compared with the traditional Brief Smell Identification Test (B-SIT). The authors suggested that the cB-SIT was similar to the B-SIT and could be administered in the diagnosis of patients with olfactory dysfunction.
In general, the Manuscript is well written and clear to understand, consequently it requires some major revisions.
Specific comments:
The enrollment of healthy subjects should be performed considering a clinical evaluation, since self-reported may be different from the medical evaluation.
Answer: Thanks for your comments. Yes, we did test the olfactory function of healthy volunteers in our previous studies. In this study, we enrolled healthy volunteers from the hospital employee. In general, they understood their medical conditions.
In the patient section, it should be “had an …" and also it should be "threshold of ...".
Answer: Thanks for your comments. We have corrected these t typographical errors.
The authors should describe how they decided on the concentration used for PEA test. Please indicate a reference for this test.
Answer: Thanks for your comments. The PEA test begins with the first trial was
presented with a -6.0 log (liquid volume/volume) concentration of PEA (the concentration of this solution was equal to 1/ 106 fold of PEA). (Assanasen P, Tunsuriyawong P, Pholpornphisit W, Chatameteekul M, Bunnag C. Smell detection threshold in Thai adults. J Med Assoc Thai. 2009;92(6):813-816.)
In the statistical analyses, authors should describe if data are normal distributed.
Answer: Thanks for your comments. In the section of statistical analyses, we added a statement to mention that our data were not normally distributed.
In the Table 2, authors should use alphabetical letter to indica the significance between groups.
Answer: Thanks for your comments. We added alphabetical letters to indicate the significance between groups in the Table 2.
In addition, a clear legend should be indicated in Table 2.
Answer: Thanks for your comments. We added a clear legend in the Table 2.
The section of test-retest reliability should be indicated in the first section of the results, since all tables descriptions should be in numerical order.
Answer: Thanks for your comments. We moved the section of test-retest reliability to the first section of the results.
Round 2
Reviewer 2 Report
Comments and Suggestions for Authors
I appreciate the revised draft of the Manuscript. My concerns were adeguately addressed by the Authors.
Comments on the Quality of English LanguageMinor revision is required